# Hot Spot Detection of Photovoltaic Module Based on Distributed Fiber Bragg Grating Sensor

**DOI:** 10.3390/s22134951

**Published:** 2022-06-30

**Authors:** Guoli Li, Fang Wang, Fei Feng, Bo Wei

**Affiliations:** 1School of Mechanical and Electrical Engineering, Jinling Institute of Technology, Nanjing 211169, China; wangfang@jit.edu.cn (F.W.); fengfei4515@126.com (F.F.); 2Wuxi Brillouin Electronic Technology Co., Ltd., Wuxi 214131, China; wb@buliyuan.com

**Keywords:** optical measurement, photovoltaic module, hot spot, fiber Bragg grating, wavelength division multiplexing

## Abstract

The hot spot effect is an important factor that affects the power generation performance and service life in the power generation process. To solve the problems of low detection efficiency, low accuracy, and difficulty of distributed hot spot detection, a hot spot detection method using a photovoltaic module based on the distributed fiber Bragg grating (FBG) sensor is proposed. The FBG sensor array was pasted on the surface of the photovoltaic panel, and the drift of the FBG reflected wavelength was demodulated by the tunable laser method, wavelength division multiplexing technology, and peak seeking algorithm. The experimental results show that the proposed method can detect the temperature of the photovoltaic panel in real time and can identify and locate the hot spot effect of the photovoltaic cell. Under the condition of no wind or light wind, the wave number and variation rule of photovoltaic module temperature value, environmental temperature value, and solar radiation power value were basically consistent. When the solar radiation power fluctuated, the fluctuation of hot spot cell temperature was greater than that of the normal photovoltaic cell. As the solar radiation power decreased to a certain value, the temperatures of all photovoltaic cells tended to be similar.

## 1. Introduction

The power generation of traditional fossil fuel thermal power is faced with environmental pollution, fuel depletion, and other problems. To achieve carbon peak and carbon neutrality, the proportion of non-fossil energy consumption will gradually increase [1]. As a green, efficient, low-cost new energy technology, photovoltaic power generation technology has been widely used in construction, agriculture, transportation, and other fields [2,3]. Photovoltaic modules are an important part of photovoltaic power generation systems. Its temperature is an important factor that affects photoelectric conversion efficiency, and the two have a negative relationship. With a 25 °C reference temperature, the actual output power drops by 0.4 W compared with the expected value when the temperature of the PV module increases by 1 °C [4]. The influence factors of the photovoltaic module temperature include the accumulation of dust on photovoltaic panels, solar radiation power, and meteorological conditions [5,6]. When a single cell in the photovoltaic module is damaged or shielded by leaves and dust, its output current becomes smaller, and both ends of the cell are reversely biased into a load of the other cells. This results in local high-temperature and hot spot effects [7,8]. The hot spot effect is a typical fault of the photovoltaic module that causes serious combustion of the whole cell module, and it seriously affects the performance and service life of the photovoltaic system [9]. Therefore, the efficient and accurate detection of photovoltaic module temperature and hot spots is of great significance to ensure the safe operation of photovoltaic power generation systems. At present, the detection methods of the temperature and hot spots for photovoltaic modules can be divided into two categories: the electrical characteristics method and the external physical characteristics method. The electrical characteristics method usually collects the output voltage, current, and other parameters by an auxiliary circuit around the photovoltaic array or photovoltaic module. Then, the detection of temperature and hot spots is conducted by a mathematical statistical model or a machine learning algorithm for data analysis. Kim et al. performed hot spot detection by AC parameters [10]. Liu and Wu et al. obtained the leakage current of hot spot cells by detecting the slope of the line segment at the step of the I-V curve of the photovoltaic module [11,12]. Jia et al. proposed a multi-sensor fault detection and location method of photovoltaic arrays based on an improved BP neural network [13]. For large-scale photovoltaic arrays, the auxiliary circuit based on the electrical characteristic detection method is a relatively large, high-cost, complex data processing method with low efficiency. Detection methods based on external physical characteristics usually achieve the detection of hot spots by the temperature distribution of the photovoltaic array. Bohorquez et al. used thermal resistance sensors or digital temperature sensors to measure the surface or back temperature of the photovoltaic module [14]. Tsanakas et al. analyzed the gray histogram and temperature contour characteristics of an infrared thermal image. The results showed that the hot spots on the photovoltaic array were related to a specific discontinuous cell. In addition, the Canny edge detection operator has been selected as a diagnostic tool to detect module-related faults leading to the hot spot heating effect [15,16]. Niazi et al. used the texture and gradient histogram features of photovoltaic module thermal images for classification, and hot spots were identified by training a machine learning algorithm [17]. Jiang et al. proposed a processing method of B-spline least-square fitting based on a gray histogram, which can suppress infrared image noise and improve the accuracy of detecting hot spots [18]. For the detection of hot spots of large-area distributed photovoltaic arrays, the efficiency of the infrared image detection method is low due to the need to collect and process a large number of thermal images. Some scholars obtained the infrared thermal image by scanning the photovoltaic array at a low altitude with a UAV and infrared thermal imager. The thermal image underwent image splicing and processing, which improved the detection efficiency to a certain extent [19,20].

The above methods have low detection efficiency and low accuracy, and they cannot be applied to detect distributed hot spots. The FBG sensor can achieve distributed monitoring with small-volume, anti-electromagnetic interference, and corrosion resistance [21,22,23,24]. It has been widely studied in distributed temperature measurement [25,26,27,28]. To solve the above problems, a hot spot detection method using a distributed FBG sensor is proposed. The FBG array is pasted on the surface of the photovoltaic panel with thermal conductive silicone grease, the shift of the reflected wavelength of FBG is demodulated by wavelength division multiplexing technology, the temperature of each measuring point is obtained, and the identification and location of hot spots are realized.

## 2. Hot Spot Effect of Photovoltaic Module and Its Influence

### 2.1. Hot Spot Effect of Photovoltaic Module

The defective areas (covered, cracked, bubble, dirt, etc.) in photovoltaic modules become loads and consume the energy generated by other areas. This results in local overheating, and then the hot spot effect of the photovoltaic module occurs, as shown in Figure 1.

The photovoltaic module is sealed by a certain number of single-chip cells in series and parallel. The circuit model of the photovoltaic cell is shown in Figure 2a. $V_{i}$ is the normal working cell output voltage, and I is the cell output current.

The I-V curvilinear equation is obtained on the basis of the equivalent circuit model of the photovoltaic cell [29]:(1)$$I=I_{\mathrm{ph}}-I_{D}\left\{\mathrm{esp}\left[\frac{q(V+IR_{s})}{nkT}\right]-1\right\}-\frac{V+IR_{s}}{R_{\mathrm{sh}}}$$
in which I is the output current, V is the output voltage, $I_{\mathrm{ph}}$ is the photogenerated current, $I_{D}$ is the diode reverse saturation current, q is the charge constant and equal to 1.6 × 10^−19^ C, $R_{s}$ is series resistance, n is the diode quality factor, k is the Boltzmann constant and the value is equal to 1.38 × 10^−23^ J/K, T is the thermodynamic temperature, and $R_{\mathrm{sh}}$ is parallel resistance.

When a photovoltaic cell has a hot spot effect, its photogenerated current decreases. As the in-string cells work at the same current intensity, the hot spot cell is reverse biased and becomes the load of the normal cells, as shown in Figure 2b, in which $V_{r}$ is the reverse bias voltage of the hot spot cell and $I_{\mathrm{st}}$ is the cell string current. When there is a hot spot cell in a photovoltaic module, its photogenerated current decreases, and the I-V characteristic curve changes, as shown in Figure 3 [11].

### 2.2. Influence Analysis of Hot Spot Effect

The power consumed by the hot spot cell is:(2)$$P=I_{\mathrm{rev}}^{2}R_{\mathrm{sh}}+I^{2}R_{s}$$
in which $I_{\mathrm{rev}}$ is the reverse leakage current. The smaller the $R_{\mathrm{sh}}$, the greater the reverse leakage current of the hot spot cell. Extremely high heat is generated and accumulated in the part with a large reverse leakage current. This results in a sharp temperature rise in this part, generating a hot spot effect. When the hot spot temperature rises to a certain extent, it destroys the surface packaging material of the module and shortens its service life. The physical structure of the single photovoltaic cell is even burnt out, and this results in permanent damage to the photovoltaic module. A fire caused by a hot spot will burn multiple photovoltaic modules or cause a large area of cell panels to be scrapped.

## 3. Hot Spot Detection of Photovoltaic Module Based on FBG

### 3.1. FBG Temperature Detection Principle

According to the coupled mode theory, The Bragg wavelength of FBG is:(3)$$\lambda_{B}=2n_{\mathrm{eff}}\Lambda$$
in which $n_{\mathrm{eff}}$ is the effective refractive index of the fiber core and Λ is the grating period. The photothermal effect caused by temperature changes the effective refractive index for bare fiber gratings without external force, and the thermal expansion coefficient changes the grating constant. The relative displacement of the Bragg wavelength caused by temperature change is:(4)$$\frac{\Delta \lambda_{B}}{\lambda_{B}}=\frac{1}{\Lambda }\frac{\partial \Lambda }{\partial T}\Delta T+\frac{1}{n_{\mathrm{eff}}}\frac{\partial n_{\mathrm{eff}}}{\partial T}\Delta T=(\alpha +\xi )\Delta T=K_{T}\Delta T$$
in which $\alpha =\frac{1}{\Lambda }{\frac{\partial \Lambda }{\partial T}}_{T}$ is the thermal expansion coefficient of optical fiber, $\xi =\frac{1}{n_{\mathrm{eff}}}\frac{\partial n_{\mathrm{eff}}}{\partial T}$ is the thermal optical coefficient of optical fiber, $K_{T}$ is the temperature sensitivity coefficient of the fiber grating relative wavelength. For fused silica fiber, α= 0.55 × 10^−6^/°C, ξ= 6.8 × 10^−6^/°C, and $K_{T}=$ 7.35 × 10^−6^/°C. When $\lambda_{B}=$ 1525 nm, the wavelength shift caused by a temperature change of 1 degree is 11.21 pm. According to Equation (4), when the optical fiber material is determined, there is a linear relationship between $\Delta \lambda_{B}$ and ΔT. The temperature change can be determined by detecting the displacement of the wavelength.

The relative displacement of the grating Bragg wavelength caused by fiber axial strain is:(5)$$\frac{\Delta \lambda_{B}}{\lambda_{B}}=(1-P_{e})\varepsilon_{z}=K_{\varepsilon }\varepsilon_{z}$$
in which $P_{e}=n_{\mathrm{eff}}^{2}[P_{12}-\upsilon (P_{11}+P_{12})]/2$, $P_{11}$ and $P_{12}$ is the elastic optic coefficient, υ is optical fiber Poisson’s ratio, and $K_{\varepsilon }$ is the relative wavelength strain sensitivity coefficient of the fiber Bragg grating.

According to Equations (4) and (5), the relationship between the change in the grating Bragg wavelength and the change in temperature and axial strain is:(6)$$\frac{\Delta \lambda_{B}}{\lambda_{B}}=(\alpha +\xi )\Delta T+(1-P_{e})\varepsilon_{z}=K_{T}\Delta T+K_{\varepsilon }\varepsilon_{z}$$

Therefore, the change in the grating Bragg wavelength is related to its axial strain and temperature. When the temperature is measured by FBG, the interference of axial strain and temperature on the grating must be considered [30].

### 3.2. Hot Spot Detection Method of Photovoltaic Module Based on FBG

The FBG array was pasted on the surface of the photovoltaic panel. The working condition of the photovoltaic module and the occurrence of hot spots were monitored by the temperature measured. The structure of the distributed FBG temperature measurement system was constructed by the wavelength division multiplexing method, as shown in Figure 4. Multiple FBGs with different Bragg wavelengths were connected in series on a transmission fiber. The wavelength of the narrow-band light source with a tunable laser output can vary within a certain range. The laser scanning step size and frequency can be controlled by the driver. The narrow-band laser was incident on the FBG array through the circulator. When the wavelength of the light source was consistent with the Bragg wavelength of an FBG, the light intensity of the reflected signal was the largest. The reflected light signal reached the photoelectric converter through the circulator, and it was converted into an electrical signal. The data processing computer collected the electrical signal, and the signal voltage peak was obtained by the peak-seeking algorithm. Then, the wavelength demodulation and positioning of the FBG were realized. By comparing the Bragg wavelength variation of each FBG sensor, the temperature of the measured photovoltaic module could be calculated. At the same time, the location of the hot spot was determined through the location of the sensor with an abnormal temperature in the FBG array. The smaller the distance between the sensors in the FBG array, the more accurate the hot spot location of the system.

## 4. Experiment and Analysis

### 4.1. Experimental Materials and Equipment

The experimental system was mainly composed of a photovoltaic panel, FBG sensors, a fiber Bragg grating demodulator, a contact digital temperature measuring instrument, an infrared thermal imager, a solar power meter, and an anemometer. The equipment model is shown in Table 1.

The length and width of the photovoltaic panel were 960 mm and 480 mm, respectively. The power of the photovoltaic power generation system was 800 W. The bandwidth of the FBG was more than 2 nm, the length of the grating area was 10 mm, and the length of the transmission fiber was 10 m. The wavelength band of the fiber Bragg grating demodulator was 1525~1565 nm, the scanning frequency was 100 Hz, and the wavelength resolution was 1 pm. The contact digital temperature measuring instrument was used to collect the surface temperature of the photovoltaic panel. The measurement results were compared with the temperature measurement results of FBG sensors. The infrared thermal imager was used to collect the infrared thermal image of the solar panel in order to observe the hot spot of the solar panel and the temperature of the surrounding area. In addition, the solar power meter and anemometer were used to measure solar power and wind speed, respectively.

The experimental platform was built according to Figure 4, as shown in Figure 5. The FBG array was composed of nine bare fiber gratings in series. The Bragg wavelengths of FBG1–FBG9 were 1530.301 nm, 1533.189 nm, 1536.113 nm, 1542.027 nm, 1544.977 nm, 1548.086 nm, 1553.949 nm, 1557.017 nm, and 1560.137, nm respectively. The FBG array was pasted onto the photovoltaic panel with thermal conductive silicone grease. The thermal conductive silicone grease used can maintain the paste state at −50~230 °C without solidifying or adhesive force. Within the temperature range of the photovoltaic module, the internal stress of thermal grease basically does not affect the Bragg wavelength of the fiber Bragg grating. In addition, the thermally conductive silicone grease can avoid uneven heating of the FBG. In the experiment, the FBG string pasted on the photovoltaic panel with thermal conductive silicone grease was arranged up and down. The upper end of the fiber was fixed with adhesive tape, and the lower end was in a free state, as shown in Figure 4. In this way, the optical fiber could be attached to the photovoltaic panel, and the FBG could be prevented from being affected by the strain deformation of the panel and other stresses. FBG1–FBG9, respectively, corresponded to points P1–P9, and the temperature at the corresponding position was measured. During the experiment, the photovoltaic panel in the working state of power generation was placed in the outdoor sunny weather. The geographic location of the experiment was latitude 31°57′8′ N, longitude 118°51′8′ E, and altitude 15 m, and the cell panel was inclined at 45°.

### 4.2. Experimental Results and Analysis

#### 4.2.1. FBG Calibration

The temperature sensitivity coefficient of the bare FBG used in the experiment was about 10 pm/°C. The temperature sensitivity coefficient and linearity of the FBG were recalibrated in the experiment. The FBG sensors to be calibrated were put into the temperature control box. The temperature of the temperature control box was set from 10~70 °C. The Bragg wavelengths of the FBGs were recorded every 10 °C. The results are shown in Figure 6. According to the calibration curve, the change in the Bragg wavelength of the sensors had a good linear relationship with the temperature. After linear fitting, the temperature sensitivities of FBG1–FBG9 sensors were, respectively, 10.50 pm/°C, 10.57 pm/°C, 10.49 pm/°C, 10.49 pm/°C, 10.50 pm/°C, 10.60 pm/°C, 10.63 pm/°C, 10.59 pm/°C, and 10.64 pm/°C. The temperature sensitivity coefficients of the nine sensors were close to each other.

#### 4.2.2. Photovoltaic Panel Temperature Measurement and Hot Spot Identification

Photovoltaic modules are often affected by shielding and dust coverage. This results in abnormal operation and even the hot spot effect. In the experiment, a plastic film with a light transmittance of 0.5 was used to cover 100% area of the cells at the P7 point to simulate the hot spot effect, as shown in Figure 5c. The surface temperature measurement experiment of the photovoltaic panel was carried out from 11:00 a.m. to 11:50 a.m. The digital thermometer (the thermometer probe was placed near the FBG sensor on the cell panel) and FBG sensors were used for temperature collection. During the measurement, it was found that the temperatures at points P1, P2, P3, P4, P5, and P6 and their variations were similar, and the temperatures at points P8 and P9 and their variations were similar. The temperature measurement results at points P1, P5, P7, and P9 are shown in Figure 7 (the digital thermometer collected the temperature every 5 min).

According to Figure 7, in most cases, the temperature at point P9 was slightly lower than that at points P1 and P5. During the experimental period, the southeast wind level II was conducive to the heat dissipation of the photovoltaic panel, especially on the point P9 side. This is the main reason for the low temperature at point P9. The temperature at point P7 on the photovoltaic panel was significantly higher, reaching 66.9 °C; this indicates that the hot spot effect occurred at this point.

The temperature stability measured by the thermometer was poor, and the temperature values were generally lower than those collected by FBG sensors. This is mainly related to the contact time between the thermometer probe and the cell panel during measurement. The contact time was too short, and the heat transfer was insufficient.

The system using the fiber Bragg grating to detect the photovoltaic module temperature and hot spots had good stability, a short response time, and good real-time performance.

Figure 8 shows the infrared thermal image of the photovoltaic panel at a certain time point. The infrared image shows the hot spot effect at point P7 and the low temperature on one side of point P9.

The FBG demodulator selected in this experiment had 8 channels, and the wavelength range of its embedded light source was 1525~1565 nm. If the bandwidth of each FBG was 2 nm, theoretically, each channel could be connected in series with 20 FBGs, and a demodulator could demodulate up to 160 FBGs using wavelength division multiplexing and space division multiplexing technology. In order to ensure the safe operation of large-scale photovoltaic panels and promptly find the hot spot fault of each photovoltaic cell, at least one fiber Bragg grating must be arranged at each cell position.

Therefore, it is necessary to increase the number of FBGs demodulated by a single demodulator or increase the number of demodulators so that the equipment investment increases accordingly. With the development of FBG multiplexing technology and demodulation technology, the number of FBGs demodulated by a single demodulator will continue to increase, and the equipment cost of using FBGs to detect the temperature of large photovoltaic arrays will gradually decrease.

#### 4.2.3. Measurement Experiment of Photovoltaic Panel Temperature and Environmental Conditions

Two FBGs were pasted on P7B and P5B on the backplane surface of the photovoltaic panel, and they were used to measure the backplane surface temperature of the photovoltaic panel. P7B and P5B corresponded to points P7 and P5, respectively. The other FBG was placed freely in the air on the front of the photovoltaic panel to measure the ambient temperature. At the same time, the solar radiation power was measured by the solar power meter. The wind speed at the experimental position was measured with an anemometer. The experiment was carried out on a cloudy day. The measurement results are shown in Figure 9.

According to Figure 9, the temperature of the back side of the photovoltaic cell with the hot spot effect was slightly higher than that of the front side, while the temperature of the photovoltaic cell with normal operation was just the opposite. As the solar radiation power decreased to a certain value, the temperatures of all photovoltaic cells tended to be similar.

Under the experimental conditions of a breeze or light wind, the number and variation law of wave peaks of the photovoltaic cell temperature value, ambient temperature value, and solar radiation power value were basically the same.

When the value of the solar radiation power was large, the ambient temperature was correspondingly high, and the temperature of the photovoltaic panel was also high, but the fluctuation range of the ambient temperature was smaller than that of the solar radiation power.

The temperature fluctuation degree of the hot spot cell caused by solar radiation power fluctuation was generally greater than that of normal photovoltaic cells. For example, at 500 s, the solar radiation power was 707 W/m^2^, the temperature at point P5 was 44.8 °C, and the temperature at point P7 was 52.7 °C. At 1100 s, the solar radiation power rose to 810 W/m^2^, the temperature at point P5 rose to 45.7 °C, while the temperature at point P7 rose to 62.1 °C.

The breeze had little effect on the temperature of the photovoltaic panel.

## 5. Conclusions

To solve the problems of the hot spot effect of photovoltaic modules and surface temperature detection of photovoltaic panels, a detection scheme that uses wavelength division multiplexing technology based on the distributed FBG sensor is proposed. In this scheme, an FBG sensor array was pasted on the surface of the photovoltaic panel, and the tunable laser method and peak-seeking algorithm were used for wavelength demodulation. Multi-point temperature measurement was performed. The detection system was built, and the measurement experiment was carried out in an outdoor environment. The results showed that:

(1) The FBG temperature sensitivity of this detection method is greater than 10 pm/°C, and the response time of the sensing probe is short. The detection system can obtain the surface temperature distribution of the photovoltaic panel in real time and can effectively identify and locate the hot spot effect of photovoltaic cells.

(2) The temperature of photovoltaic cells with a hot spot effect is significantly higher than that of normal working cells. The temperature of the upper wind side of the photovoltaic panel is slightly lower than that of the lower wind side due to the influence of air cooling.

(3) When the value of solar radiation power is large, the ambient temperature is correspondingly high, and the temperature of the photovoltaic panel is also high. As the solar radiation power decreases to a certain value, the temperature of all photovoltaic cells tends to be similar.

(4) The breeze has little effect on the temperature of the photovoltaic panel. Under the experimental conditions of a breeze or light wind, the number and variation law of the wave peaks of the photovoltaic module temperature value, ambient temperature value, and solar radiation power value are basically the same.

(5) The temperature fluctuation of hot spot cells caused by solar radiation power fluctuation is generally greater than that of normal photovoltaic cells.

The distributed FBG sensor network has good real-time performance, corrosion resistance, and electromagnetic interference resistance. It can achieve multi-point and large-area temperature measurement and positioning. It has good application prospects in the field of photovoltaic panel temperature monitoring.

## Figures and Tables

**Figure 1 sensors-22-04951-f001:**
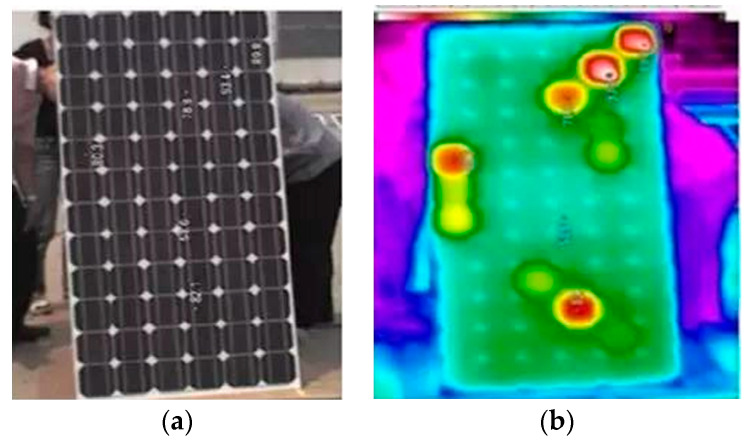
Hot spot effect of photovoltaic module: (**a**) photovoltaic module; (**b**) infrared image of the photovoltaic module.

**Figure 2 sensors-22-04951-f002:**
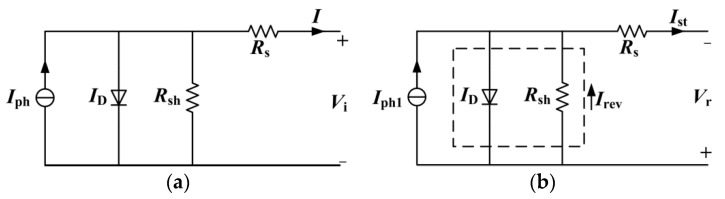
Photovoltaic cell single-diode model: (**a**) normal photovoltaic cell; (**b**) hot spot cell.

**Figure 3 sensors-22-04951-f003:**
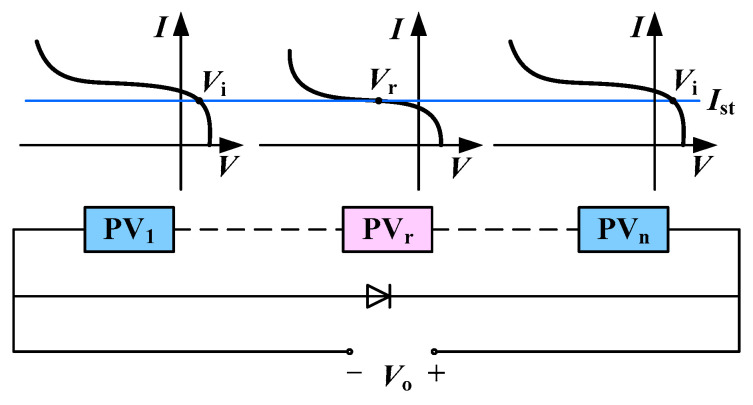
Schematic diagram of photovoltaic cell string and I-V characteristic curve.

**Figure 4 sensors-22-04951-f004:**
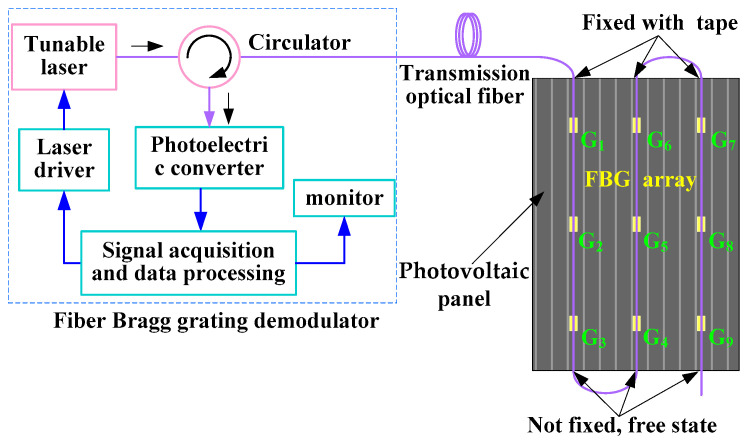
Structure diagram of distributed FBG sensor system.

**Figure 5 sensors-22-04951-f005:**
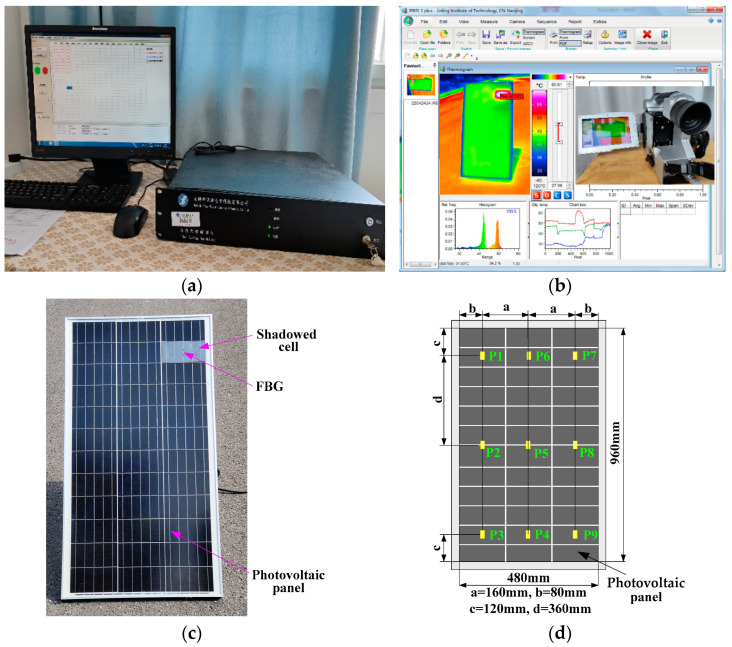
Experimental platform: (**a**) FBG demodulator; (**b**) infrared camera and image processing software; (**c**) photovoltaic panel and FBG layout; (**d**) temperature measurement location.

**Figure 6 sensors-22-04951-f006:**
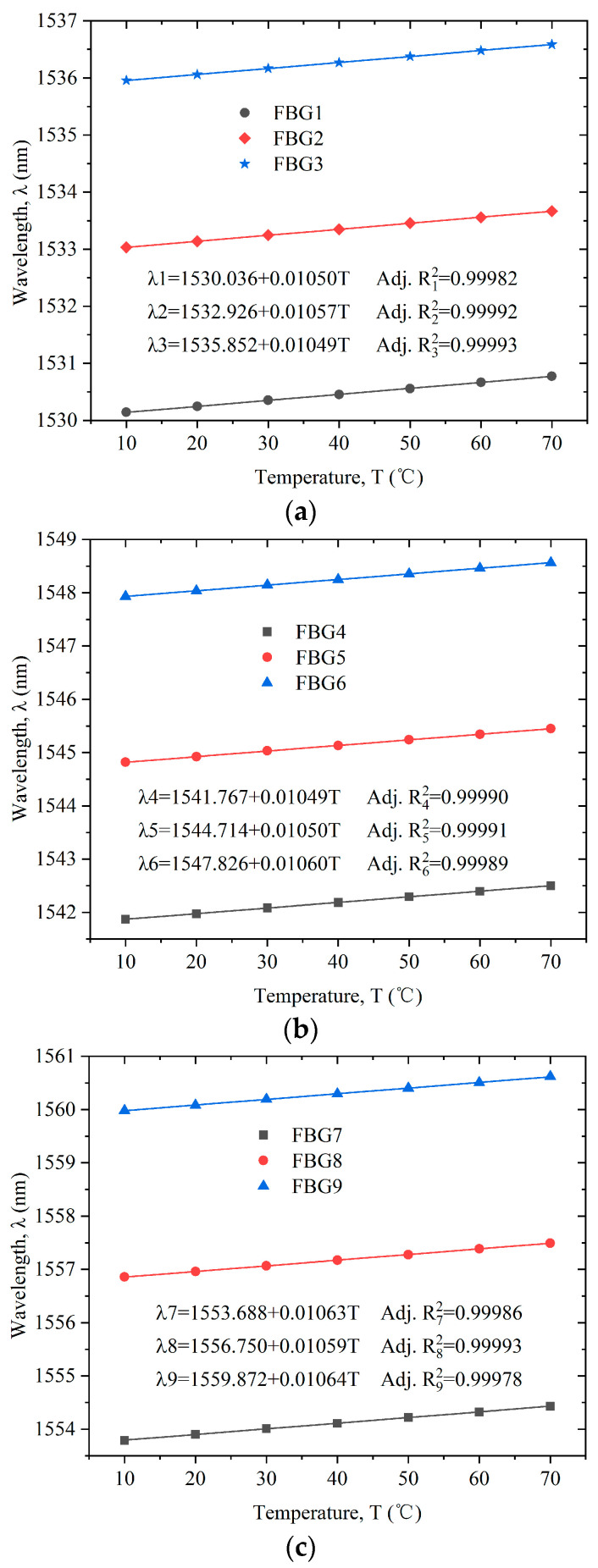
Calibration curves of the FBG sensors: (**a**) FBG1–FBG3; (**b**) FBG4–FBG6; (**c**) FBG7–FBG9.

**Figure 7 sensors-22-04951-f007:**
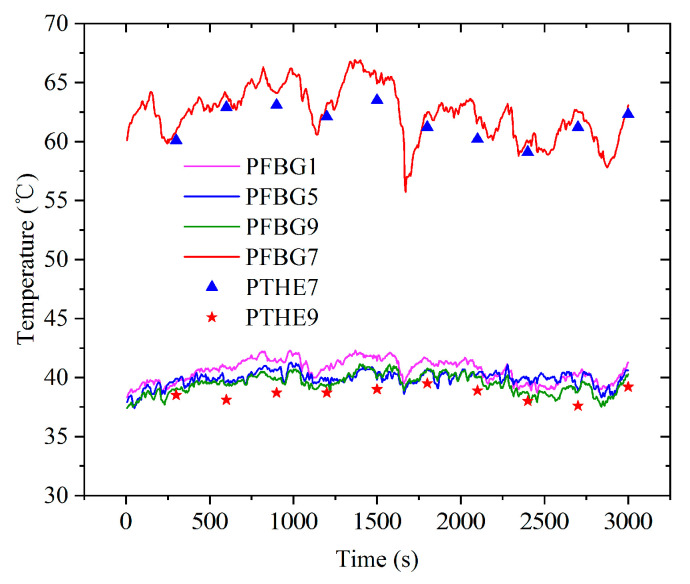
Temperature measurement results of photovoltaic panel surface.

**Figure 8 sensors-22-04951-f008:**
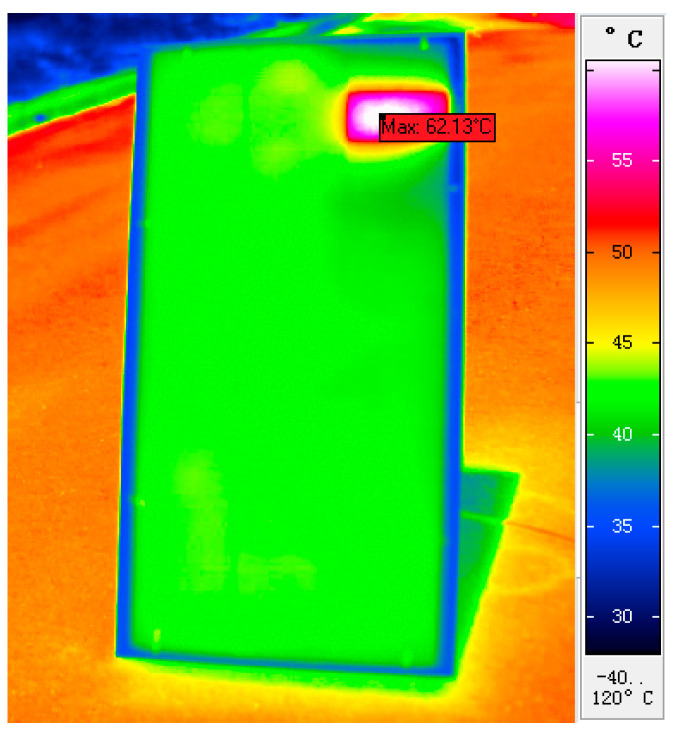
Infrared thermal image of photovoltaic panel with hot spot effect.

**Figure 9 sensors-22-04951-f009:**
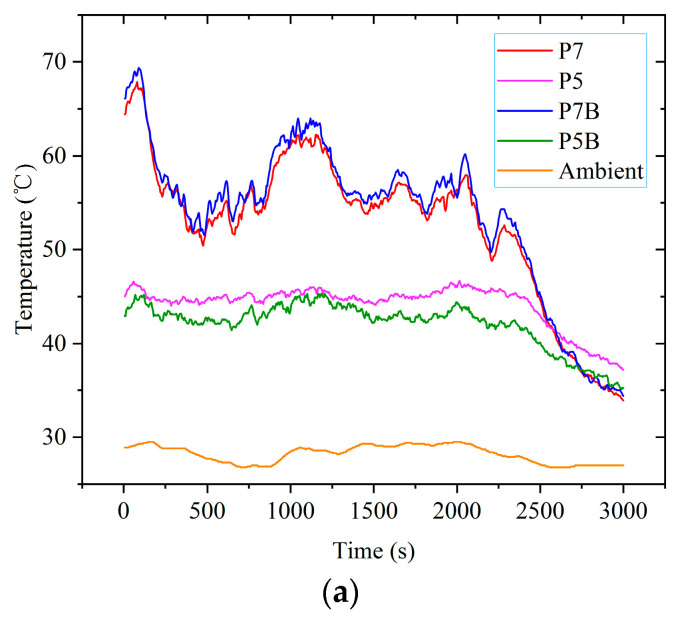
Photovoltaic panel surface temperature and environmental conditions: (**a**) temperature of photovoltaic panel and ambient; (**b**) solar radiation power; (**c**) wind speed.

**Table 1 sensors-22-04951-t001:** Experimental equipment.

Equipment	Model	Manufacturer
Photovoltaic panel	Polycrystalline, 960 mm × 480 mm	——
FBG demodulator	BLY-FBG-5S, 1525~1565 nm	China Wuxi Brillouin Electronic Technology Co., Ltd. (Wuxi, China)
Digital thermometer	TASI-TA612C	China Suzhou TASI Electronic Industry Co., Ltd. (Suzhou, China)
Infrared thermal imager	VarioCAM^®^HD inspect 980	Germany InfraTec (Dresden, Germany)
Solar power meter	TES-1333R	China TES Electrical Electronic Corp (Taibei, China)
Anemometer	Benetech GM8907	China Shenzhen Jumaoyuan Science And Technology Co., Ltd. (Shenzhen, China)

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
