# Peer review of "Hot Spot Detection of Photovoltaic Module Based on Distributed Fiber Bragg Grating Sensor"

_sensors, 2022, doi:10.3390/s22134951_

Round 1

Reviewer 1 Report

The authors presented a research that describes a method for continuous monitoring of the PV Modules based on utilization of Bragg gratings. The article consist of 5 sections (introduction, hot spot effect on PV, System design based on FBGs, Experiments and Analysis and conclusions).

The article is in general well written. The introduction section describes a significant number of research papers that discusses the problem described in the research. However, I have several comments to the text of the article:

11)    Authors stated that they have used a conductive silicone grease. However, I have doubts about the material. If the temperature will rise, I believe that it will affect the internal stresses of the silicon, and thus, the Bragg wavelength will be affected. Could authors discuss on that? I believe that calibration process described in 4.2.1 is related to the bare fibers, not attached to the panels.

22)    I have also doubts about other effects (such as described in 4.2.3 influence of the wind speed). It also may affect the stress of the PV module and in turn Bragg wavelength will be “mechanically” affected. I believe that this issue needs to be discussed and solved. For instance, a Bragg grating sensor that eliminates the cross-sensitivity can be used (see: https://doi.org/10.1364/OL.390784). I am strongly recommending to authors discussion on that issue.

33)    Authors should work on the issue of the nomenclature used in the research. For instance line 134 Coupled Membrane Theory -> Coupled Mode Theory; line 136 grid period -> grating period. Authors should review in details all the text of the publication.

Nonetheless, the idea presented in this research may be found very interesting for numerous of researchers. Thus, when (1-3) points of this review will be included, I am recommending this paper for publication.

Author Response

Dear reviewer,

Thank you for your valuable and constructive comments and suggestions for our manuscript. Those are all valuable and very helpful for revising and improving our paper, as well as having the important guiding significance to our research. We have studied comments/suggestions very carefully and made revisions which we hope it meet with approval. The main corrections in the paper and the responses to the reviewer’s comments are provided in a point-by-point manner as following.

Kind regards,

Authors

Manuscript Title: Hot Spot Detection of Photovoltaic Module Based on Distributed Fiber Bragg Grating Sensor

Manuscript ID: sensors-1779287

Comment #1: Authors stated that they have used a conductive silicone grease. However, I have doubts about the material. If the temperature will rise, I believe that it will affect the internal stresses of the silicon, and thus, the Bragg wavelength will be affected. Could authors discuss on that? I believe that calibration process described in 4.2.1 is related to the bare fibers, not attached to the panels.

Response:

We thank the reviewer for bringing this to our attention. The cross-sensitivity of temperature and stress of FBG proposed by the reviewer is a key problem in FBG temperature measurement. With the increase of temperature, the internal stress of thermal conductive silicone grease really needs to be studied.

According to the reviewer's comments, we further consult the relevant information. The thermal conductive silicone grease can maintain the grease state at the temperature of -50 °C ~230 °C, without solidify and adhesive force. Within the temperature range of photovoltaic module, the internal stress of thermal grease is stable, which basically does not affect the Bragg wavelength of fiber Bragg grating.

Therefore, the FBG can avoid stress interference by pasting the fiber onto the photovoltaic panel with thermal conductive silicone grease. According to the comments of reviewer, the following contents are added to the third paragraph of Section 4.1 of the paper for further explanation: "The thermal conductive silicone grease used can maintain the paste state at -50 °C ~230 °C, without solidify and adhesive force. Within the temperature range of photovoltaic module, the internal stress of thermal grease basically does not affect the Bragg wavelength of fiber Bragg grating. In addition, the thermal conductive silicone grease can also avoid uneven heating of FBG".

Comment #2:  I have also doubts about other effects (such as described in 4.2.3 influence of the wind speed). It also may affect the stress of the PV module and in turn Bragg wavelength will be “mechanically” affected. I believe that this issue needs to be discussed and solved. For instance, a Bragg grating sensor that eliminates the cross-sensitivity can be used (see: https://doi.org/10.1364/OL.390784). I am strongly recommending to authors discussion on that issue.

Response: We thank the reviewer for her/his comments and suggestion to browse the relevant literature. When the fiber Bragg grating is pasted on the photovoltaic panel, it is really necessary to pay attention to the influence of mechanical stress. Now the following content has been added in Section 4.1 of the paper for further explanation: "In the experiment, the FBG string pasted on the photovoltaic panel with thermal conductive silicone grease is arranged up and down. The upper end of the fiber is fixed with adhesive tape, and the lower end is in a free state, as shown in Figure 4. In this way, the optical fiber can be attached to the photovoltaic panel, and the FBG can be prevented from being affected by the strain deformation of the panel and other stresses". Figure 4 has also been modified.

we have carefully studied the literature on the cross-sensitivity of FBG suggested by the reviewer, listed this literature as a reference, and marked it in the last part of Section 3.1 of the paper.

  1. Konrad Markowski; Piotr Araszkiewicz; Juliusz Bojarczuk; Krzysztof Perlicki. High-sensitivity chirped tapered fiber-Bragg-grating-based Fabry–Perot cavity for strain measurements. Lett.45, 2020, 10,2838-2841. https://doi.org/10.1364/OL.390784

Comment #3: Authors should work on the issue of the nomenclature used in the research. For instance line 134 Coupled Membrane Theory -> Coupled Mode Theory; line 136 grid period -> grating period. Authors should review in details all the text of the publication. 

Response: We kindly thank the reviewer for finding our omissions. These problems found by the reviewer are very important to this paper. We recheck the full text and correct the incorrect description of terms, as mentioned by the reviewer: coupled membrane theory→coupled mode theory; grid period → grating period.

Reviewer 2 Report

This paper proposes a hot spot detection method of photovoltaic module based on distributed FBG sensors. It is a feasible and practical application, and can effectively solve the hot spot detection problem of low accuracy, low efficiency, and difficulty of photovoltaic module. The experimental conditions are introduced in detail. The results show that the detection method based on FBG sensors is feasible. A key issue to notice is as follows:

The content of section 2 is suggested to be deleted or edited again in section 1. The hot spot effect of photovoltaic module and its influence should be briefly introduced in the introduction rather than in the main body.

Author Response

Dear reviewer,

Thank you for your valuable and constructive comments and suggestions for our manuscript. Those are all valuable and very helpful for revising and improving our paper, as well as having the important guiding significance to our research. We have studied comments/suggestions very carefully and made revisions which we hope it meet with approval. The main corrections in the paper and the responses to the reviewer’s comments are provided in a point-by-point manner as following.

Kind regards,

Authors

Manuscript Title: Hot Spot Detection of Photovoltaic Module Based on Distributed Fiber Bragg Grating Sensor

Manuscript ID: sensors-1779287

Comment: The hot spot effect of photovoltaic module and its influence should be briefly introduced in the introduction rather than in the main body.

Response:

We thank the reviewer for her/his careful reading of the manuscript. The main content of this paper is the detection of hot spot of photovoltaic module based on fiber Bragg grating sensors. Therefore, Section 2 is devoted to the principle of hot spot and its influence. The discussion about the causes of the hot spot effect in Section 2 of the paper is indeed somewhat repeated with that in the introduction.

According to the reviewers' comments, we deleted the content in Section 2: "There are two main causes for photovoltaic hot spot. First, photovoltaic cells become the load of other cells because they cannot generate power due to shielding, resulting in internal power consumption and hot spots. Second, If there are defects in the photovoltaic module or part of the cell itself, or the module is broken due to negligence in the process of handling and assembly, hot spots will happen because the module can not work normally".

Thus, the Introduction mainly introduces the causes of hot spots, and Section 2 mainly introduces the mechanism of hot spot. We thank you again for your suggestion.

Reviewer 3 Report

A method for monitoring critical temperature conditions of photovoltaic panels is presented. Its effectiveness is assessed through experimental measurements. The following comments should be addressed.

1.       Apparently, the number of FBGs that can be monitored by a single demodulator is limited by the by the available range for the wavelengths. This may be an issue for monitoring large arrays of solar panels, since a large number of demodulators would be also required. Some considerations on these aspects should be included in the paper, reasonably in Section 4 or 5.

2.       How fast can be the temperature transients that lead to hot spot phenomena? Is real-time monitoring strictly required? If not, may a proper scheduling of the acquisitions be adopted for increasing the number of FBGs monitored by a single demodulator? Some considerations on these aspects should be included in the paper.

3.       In the introduction, some recent works dealing with the use of FBGs for solar energy related applications (e.g. Ref. [a], listed below) may be added to the References.
[a] Falcetelli, F.; Martini, A.; Di Sante, R.; Troncossi, M. Strain Modal Testing with Fiber Bragg Gratings for Automotive Applications. Sensors 2022, 22, 946. https://doi.org/10.3390/s22030946

4.       In Figure 4, some of the subscripts related to the FBGs are out of place (namely, G_3 and G_5).

5.       There is a repetition in the Abstract (“of light wind or light wind”). Please, check.

Author Response

Dear reviewer,

Thank you for your valuable and constructive comments and suggestions for our manuscript. Those are all valuable and very helpful for revising and improving our paper, as well as having the important guiding significance to our research. We have studied comments/suggestions very carefully and made revisions which we hope it meet with approval. The main corrections in the paper and the responses to the reviewer’s comments are provided in a point-by-point manner as following.

Kind regards,

Authors

Manuscript Title: Hot Spot Detection of Photovoltaic Module Based on Distributed Fiber Bragg Grating Sensor

Manuscript ID: sensors-1779287

Comment #1: Apparently, the number of FBGs that can be monitored by a single demodulator is limited by the available range for the wavelengths. This may be an issue for monitoring large arrays of solar panels, since a large number of demodulators would be also required. Some considerations on these aspects should be included in the paper, reasonably in Section 4 or 5.

Response:

We thank the reviewer for this comment. The equipment investment problem of large-scale photovoltaic array inspection project based on FBG is really worth considering.

According to the reviewer's comments, the following content is now added at the end of Section 4.2.2 (The following also discusses the issue of Comment #2):

" The FBG demodulator selected in this experiment has 8 channels, and the wavelength range of its embedded light source is 1525 nm~1565 nm. If the bandwidth of each FBG is 2nm, theoretically each channel can be connected in series with 20 FBGs, and a demodulator can demodulate up to 160 FBGs using wavelength division multiplexing and space division multiplexing technology. In order to ensure the safe operation of large-scale photovoltaic panels and timely find the hot spot fault of each photovoltaic cell, at least one fiber Bragg grating shall be arranged at each cell position.

Therefore, it is necessary to increase the number of FBGs demodulated by a single demodulator or increase the number of demodulators, so that the equipment investment will increase accordingly. With the development of FBG multiplexing technology and demodulation technology, the number of FBGs demodulated by a single demodulator will continue to increase, and the equipment cost of using FBG to detect the temperature of large photovoltaic arrays will gradually decrease."

Comment #2:  How fast can be the temperature transients that lead to hot spot phenomena? Is real-time monitoring strictly required? If not, may a proper scheduling of the acquisitions be adopted for increasing the number of FBGs monitored by a single demodulator? Some considerations on these aspects should be included in the paper.

Response: We thank the reviewer for bringing this to our attention. The temperature rise caused by the hot spot changes rapidly and needs to be detected in real time. In order to ensure the safe and stable operation of the photovoltaic power generation system, it is necessary to detect the temperature and hot spot fault of each cell in the photovoltaic array in real time. To this end, at least one FBG needs to be arranged at each cell to detect its temperature, so it is necessary to increase the number of FBG multiplexing of a single demodulator or increase the number of demodulators, and the equipment investment will increase accordingly. This issue is also discussed in the response to the previous question.

Comment #3: In the introduction, some recent works dealing with the use of FBGs for solar energy related applications (e.g. Ref. [a], listed below) may be added to the References.
[a] Falcetelli, F.; Martini, A.; Di Sante, R.; Troncossi, M. Strain Modal Testing with Fiber Bragg Gratings for Automotive Applications. Sensors 202222, 946. https://doi.org/10.3390/s22030946

Response: Thank you for this comment. According to the reviewer's comments, we consult the relevant literatures, and carefully studied the literature suggested by the reviewer, and finally added it to the references.

  1. Falcetelli, F.; Martini, A.; Di Sante, R.; Troncossi, M. Strain Modal Testing with Fiber Bragg Gratings for Automotive Applications. Sensors202222, 946. https://doi.org/10.3390/s22030946

Comment #4: In Figure 4, some of the subscripts related to the FBGs are out of place (namely, G_3 and G_5).

Response: We are very grateful to the reviewer for finding our omissions. We have checked the manuscript and corrected the problems in Figure 4.

Comment #5: There is a repetition in the Abstract (“of light wind or light wind”). Please, check.

Response: Once again, we thank the reviewer for finding our omission. We have checked the manuscript and corrected the problems in the Abstract: "of light wind or light wind""of no wind or light wind".

Round 2

Reviewer 1 Report

Thank you - all my doubts have been discussed. In My opinion manuscript is ready for publication